# Individual Sprint Force-Velocity Profile Adaptations to In-Season Assisted and Resisted Velocity-Based Training in Professional Rugby

**DOI:** 10.3390/sports8050074

**Published:** 2020-05-25

**Authors:** Johan Lahti, Pedro Jiménez-Reyes, Matt R. Cross, Pierre Samozino, Patrick Chassaing, Benjamin Simond-Cote, Juha P. Ahtiainen, Jean-Benoit Morin

**Affiliations:** 1Le Laboratoire Motricité Humaine Expertise Sport Santé, Université Cote d’Azur, 06200 Nice, France; patrickchassaing@live.fr (P.C.); jeanbenoitmorin@gmail.com (J.-B.M.); 2Centre for Sport Studies, Rey Juan Carlos University, 28933 Madrid, Spain; peterjr49@hotmail.com; 3Laboratoire Interuniversitaire de Biologie de la Motricité, Université Savoie-Mont Blanc, EA 7424, F-73000 Chambéry, France; cross.matt.r@gmail.com (M.R.C.); pierre.samozino@univ-smb.fr (P.S.); 4Sports Performance Research Institute New Zealand (SPRINZ), Auckland University of Technology, Auckland 1010, New Zealand; 5FC Grenoble Rugby, 38000 Grenoble, France; benjamin.simondcote@hotmail.fr; 6Faculty of Sport and Health Sciences, Neuromuscular Research Center, University of Jyväskylä, 40014 Jyväskylä, Finland; juha.ahtiainen@jyu.fi

**Keywords:** sprinting, resistance training, overspeed, horizontal force, velocity-based training

## Abstract

We tested the hypothesis that the degree of adaptation to highly focused sprint training at opposite ends of the sprint Force-Velocity (FV) spectrum would be associated with initial sprint FV profile in rugby athletes. Training-induced changes in sprint FV profiles were computed before and after an eight-week in-season resisted or assisted sprint training protocol, including a three-week taper. Professional male rugby players (age: 18.9 ± 1.0 years; body height: 1.9 ± 0.0 m; body mass: 88.3 ± 10.0 kg) were divided into two groups based on their initial sprint FV profiles: 1) Heavy sled training (RESISTED, N = 9, velocity loss 70–80%), and 2) assisted acceleration training (ASSISTED, N = 12, velocity increase 5–10%). A total of 16 athletes were able to finish all required measurements and sessions. According to the hypothesis, a significant correlation was found between initial sprint FV profile and relative change in sprint FV profile (RESISTED: r = −0.95, p < 0.01, ASSISTED: r = −0.79, p < 0.01). This study showed that initial FV properties influence the degree of mechanical response when training at different ends of the FV spectrum. Practitioners should consider utilizing the sprint FV profile to improve the individual effectiveness of resisted and assisted sprint training programs in high-level rugby athletes.

## 1. Introduction

Sprint acceleration is a key physical performance parameter that has been shown to distinguish high- and low-level athletes within rugby [1,2]. An athlete’s acceleration capacity is determined largely by the ability to produce and maintain a high horizontal force component in relation to the sum force vector with increasing velocity [3,4]. This ability is well described through the sprinting force-velocity (FV) relationship. Thus, it is logical that targeting the FV relationship within training is of interest. Although it has been common practice to refer periodization concepts surrounding FV relationships for decades [5,6], intervention studies quantifying the FV relationship within sprinting to test its adaptability to specific training modalities are sparse [7,8,9]. Evidently, this is partly due to that fact that there has only recently been developments in the accessibility of such data [10]. 

In sprinting, the FV relationship provides information regarding the production of force capacity at low velocities (theoretical maximum force (*F0*)) and at high velocities (theoretical maximum velocity (*v0*)). The balance and contribution of these variables have been shown to be predictive of sprinting performance at various distances [11]. Because force expression at different velocities is underlined by multiple neurophysiological properties [12], targeting specific overload in different zones of the FV spectrum could lead to controlled and possibly isolated shifts in the FV spectrum. This is considered to take place either with a bias toward the force end or velocity end, or a balanced shift, which would likely depend on the training method and the individual’s current FV relationship status [5]. This could be a valuable avenue of chasing sport-/situation-specific adaptations. In an effort to improve the targeting of appropriate training stimuli for athletes, training according to an individual’s FV relationship has shown promise, albeit with a focus on jumping performance [13]. Based on the initial FV properties of the athlete, the goal is to change the athlete’s FV relationship into a desired direction for performance improvements using training modalities with different loading schemes. This would lead the FV relationship to become more velocity or force oriented or lead a more balanced change. Thus, by taking the athletes initial FV properties into consideration, a group approach can be replaced by an individualized approach, which can lead to a higher performance transfer for the athlete population. 

In this regard, training using horizontal resistance, such as sleds, is used by practitioners to favor the development of the horizontal force component across the FV spectrum. Thus, greater loads are used to aid development in force properties at low velocities and lighter loads at higher velocities, changing the FV relationship with a bias toward force orientation [14,15,16]. On the other hand, assisted sprinting, such as using elastic cords, bungees, or a pulley for assistance, targets the opposite end of the spectrum and potentially increases velocity orientation of the FV profile (improving the horizontal force component at high velocities) [16,17]. Furthermore, assisted training can even surpass the FV spectrum, providing supramaximal velocity stimuli [16]. These two training modalities provide a unique and potentially valuable method of overloading distinct capabilities and consequently providing specific mechanical adaptations via selecting loading (or assistance) depending on the current profile and performance needs of the athlete. Sprint training utilizing the sprint FV profile combined with velocity-based loading has only been discussed in literature as a missing link for training optimization [8]. Thus, it is still unknown whether changes in an athlete’s sprint FV profile can be effectively targeted via horizontally focused velocity-based training at different ends of the FV spectrum, and whether targeted changes depend on an athlete’s initial FV properties. 

Furthermore, optimizing the length between training completion and post-testing is of additional interest. It is common for post-testing to be done the week after program completion, even though it has been shown on both a group and individual level that peaking response can be further delayed [18]. Therefore, providing more than one week of post-testing might aid the interpretation of the value of the training modalities. 

Therefore, our aim was to quantify the influence of initial sprint FV properties on changes in individual sprint FV profiles using training at different ends of the sprint FV spectrum and to demonstrate individual peaking responses. Specifically, changes were targeted using horizontally oriented training modalities; resisted and assisted training, standardized by velocity. Therefore, our aim was to test the initial potential for utility of contrasting horizontally oriented training modalities while producing shifts in the sprint FV profile. Our hypothesis was that in professional rugby, changes in the sprint FV profile caused by sprint modalities on opposite ends of the sprint FV spectrum would be highly associated with the athlete’s initial sprint FV profile and that most athletes would peak within the last two weeks of post-testing.

## 2. Materials and Methods

### 2.1. Participants

Twenty-one male professional and semiprofessional rugby athletes from Grenoble, France, volunteered to participate in the study using convenience sampling. All athletes had experience in sled training across multiple seasons, but not in assisted training initially, and had over 10 years of competitive practice in rugby. The inclusion criteria included being an active and healthy rugby player within the team, while exclusion criteria included completing less than 12 out of 16 training sessions and completing less than all testing sessions. Due to scheduling and injury issues, 5 out of 21 athletes could not perform the minimum required sessions and/or the post-testing and were removed from the study. Specifically, three of the dropouts were due to scheduling issues (international games, RESISTED N = 1, ASSISTED N = 2) that led to the players missing the post-testing. Two athletes dropped out due to injuries (RESISTED N = 2). According to the team physiotherapist, one was a contact injury sustained in a match, and the other was a reaggravated chronic foot problem reported in the first weeks of training, possibly aggravated by the sled protocol. Consequently, the final analysis was performed on 16 athletes. Six athletes completed heavy resisted training (age: 19 ± 0.3 years; height: 1.83 ± 0.1 m; mass: 91.4 ± 15.3 kg), and ten athletes performed the assisted training (age: 20 ± 1 years; height: 1.9 ± 0.1 m; mass: 94.4 ± 9.1 kg). The athletes were in the initial in-season phase, trained four days, and competed once per week, which included strength and plyometric training once a week (Table 1). The participants were verbally informed on the details or the study and voluntarily signed informed consent documents before participation. The testing procedures were approved by the local ethics review board LAMHESS (#2016-08) and were performed according to Declaration of Helsinki.

### 2.2. Study Design

The aim for both groups was to achieve a minimum of 12 sessions out of 16 available sessions within 8 weeks. The first four weeks involved three resisted or assisted sprints with one to two free sprints twice per week. The last four weeks involved four resisted or assisted sprints with one free sprint twice per week (Figure 1). All training and testing sessions were completed inside on artificial turf. All athletes were harnessed at their waist, while the RESISTED cohort used their own custom-made sprint sleds, and ASSISTED group utilized robotic pulley device to provide isotonic assistance (1080 Sprint, 1080 Motion, Lidingö, Sweden). To standardize the stimuli within groups, a velocity-based training approach was utilized, where all athletes used a load that adapted their velocity to the desired threshold, corresponding to a specific section of the FV relationship. The 0–10-m (RESISTED) vs. 0–20-m (ASSISTED) sprint structure was used to roughly standardize time under tension (ASSISTED: ~3 s, RESISTED: ~5 s) while reducing the risk for sprint-related injuries within the season. Training was supervised by team strength and conditioning coaches and completed before technical and/or tactical training. Pre-training warm-up (~15 min) included light running, dynamic full body stretches, muscle and dynamic movement pattern activation, and low-to-high intensity sprint exercises. The between-sprint rest was 2–3 min. The sled load and the pulley assistance were chosen with the aim to safely maximize divergence between the two groups. Specifically, the stimuli was based on a combination of our observations and relevant literature [7,8,9,17] to provide a stimulus for developing disparate sections of the force-velocity spectrum. For the sled, the load had to be high enough to be considered as maximal strength training without limiting dynamic movement or inducing clear torso rotation. These criteria produced a targeted maximal velocity reduction of 70–80% of *v0* (roughly 75–85% from Vmax). This is also why the pulley system could not be used for resistance, as its maximal resistance capacity was not high enough for such a velocity drop. Although the goal in assisted acceleration sprinting was to stimulate the entire acceleration, safety concerns dictated that assistance be standardized by the increase in velocity at the end of the sprint where the injury risk is likely highest. Therefore, a combination of our observations involving technique analysis and previous literature resulted in a maximal velocity increase target of 105–110% [16,19]. Since the pulley system provides digital feedback, velocities were verified on a weekly basis. Finally, to improve the understanding of individual supercompensation behavior after a sprinting intervention, post-testing was completed over a three-week period in the form of a ‘exponential’ taper [20]. This was done by reducing the modality specific volume completely while maintaining completing sprint FV tests (2 × 30-m free sprints) on a standardized day each week.

### 2.3. Group Allocation

To improve control of the research question, athletes were ordered from lowest to highest via sprint FV profiles derived during familiarization. Athletes were allocated to both groups to balance variance, with two additional athletes allocated to the ASSISTED group due to athlete and coaching staff preference (RESISTED N = 9: FV slope: −0.85 ± 0.09 vs. ASSISTED N = 12: FV slope: −0.87 ± 0.08, p = 0.57). The experimental groups initiating the program consisted of 9 participants in the RESISTED group and 12 in the ASSISTED group. However, there were a total of 5 dropouts (see participant section for more information), and a total of 16 athletes finished all of the required training and testing sessions (RESISTED: N = 6, ASSISTED: N: 10). The dropouts did not substantially affect the high sprint FV profile homogeneity between groups (RESISTED FV slope: −0.85 ± 0.09, ASSISTED FV slope: −0.86 ± 0.09, p = 0.83).

### 2.4. Familiarization and FV Profile Tests

Familiarization was initiated by the entire cohort three weeks before the training intervention and was combined with sprint FV profile tests (2 × 30-m sprints) on the last two weeks. Because no data was available before the first familiarization session, a load of 100% of BM (2 × 10-m sprints) was selected. In the second week, the data from the FV profiling was used to place athletes into their training groups and provide modality specific familiarization (i.e., assistance or resistance). Heavy sled familiarization was combined with assisted sprint training familiarization on the first week of familiarization (2 × 20-m sprints). The final familiarization session was combined with load-velocity testing (see the following sections) for the RESISTED group and assisted sprints for the ASSISTED group. The assistance set for the ASSISTED group was initially 7.5% of BM. Within the three weeks of familiarization, the assistance set was manipulated to reach speeds of 5–10% of maximal velocity confirmed by the isotonic pulley device.

### 2.5. Testing Procedures and Data Analysis 

Sprint FV profiling tests. After warm-up, athletes performed two 30-m maximal sprints from a standing staggered stance start, with 3 min of passive recovery between sprints. For the best time trial (highest F*0*), sprint performance (split times 0-5 and 0-20 m) and mechanical outputs were computed pre- and post-training using a validated field method measured with a radar device (Stalker ATS Pro II, Applied Concepts, Richardson, TX, USA), as reported previously [10,21,22]. Briefly, this computation method is based on a macroscopic inverse dynamics’ analysis of the center-of-mass motion. Raw velocity-time data were fitted by an exponential function. Instantaneous velocity data was then combined with system mass (body mass) and aerodynamic friction to compute the net horizontal anteroposterior ground reaction force. Individual linear sprint force-velocity (FV) profiles were then extrapolated to calculate relative theoretical maximal force (*F0:* N kg^−1^), velocity (*v0*: m/s) capabilities, and peak power (Pmax: W kg^−1^) capabilities in the anteroposterior direction. Pmax is an approximate measurement of “maximal power,” which is only derived from the forward running velocity and the anterior-posterior force, which should technically be called a pseudo-power [23]. We use the term “Pmax” to describe maximal power output in this study. The sprint with the highest peak power was utilized for analysis. 

Load-velocity tests. Commonly, load is either standardized according to % of BM or a velocity-based training approach [14,15]. To control for both internal (relative strength) and external (friction) factors and improve individualization of load, determining loading parameters according to the velocity-based training approach is likely a more effective approach [24]. Load-velocity tests were completed under one unloaded and three loaded conditions with one sprint per load (50%, 75%, 100% of BM) for the RESISTED group outlined in previous literature [25]. The load-velocity data was then fit with a least-square linear regression to generate an individualized load-velocity profile for each athlete. Thereafter, the load corresponding to a 75% velocity loss of maximal velocity was calculated. Logistical limitations of the radar device resulted in sled velocity not being confirmed in the actual training weeks. The interindividual coefficient of variation (CV%) for predicting 75% velocity reduction was 6.31% (CL95%: 3.90–8.80) from our pilot data within a similar population, which was deemed acceptable considering the substantial magnitudes of sled loads used (range: 95–120% of BM). The error was expected to be similar in the rugby cohort due to surface and sled similarities and highly accurate linear fits on the load-velocity profile (r^2^ = 0.98 ± 0.01). Therefore, no corrections were made to loads. 

### 2.6. Statistical Analysis

Normality of the data was ensured using Shapiro–Wilk test of normality. Pre- and post-testing was performed on multiple occasions. From the two weeks of pre-testing, the testing day, including the best sprint performance (F*0*), was used for statistical analysis. From the three weeks of post-testing, the largest change was utilized for statistical analysis, i.e., if all post-testing values were negative in nature, then the lowest negative value was utilized. Levene’s test was used to examine the homogeneity of variance for variables of interest (*F0*, *v0*, Pmax, sprint FV profile slope, 5-m and 20-m split times). Two-tailed dependent and independent t-tests were used to examine within-group differences. Between-group differences were assessed by a one-way ANCOVA with Bonferroni post-hoc, which included controlling for the effect of initial sprint mechanical and performance variables (covariate in ANCOVA model). All of the above-mentioned tests were performed using SPSS software version 22.0 (SPSS Inc., Chicago, IL, USA). Effect sizes (ES) were calculated using pooled SD from the two groups with 95% confidence limits using a custom spreadsheet [26], allowing interpretation of our data against Hopkins’ benchmarks to assign small (≥0.2), moderate (≥0.6), large (≥1.2) effects [27]. Pearson correlation coefficients were calculated between initial sprint FV profile (*-F0/v0*), *F0* and *v0*, and % changes in all monitored variables. In an effort to account for normal fluctuations in athletes’ weekly sprint performance during the season, minimum detectable change (MDC) at a 95% confidence interval was calculated as Typical Error (TE) × 1.96 √ 2 from the difference in best performance sprint FV profile variables (*F0, v0*, Profile slope, 5-m and 20-m time) completed during pre-test week 1 and 2. The MDC% was defined as (MDC/X̅) × 100 [28]. Alpha was set at p < 0.05, and Bonferroni adjustments were made for multiple comparisons (0.05/6 = 0.008) [29]. Descriptive data are presented as mean ± standard deviation (SD). 

## 3. Results

The RESISTED group completed an average of 12.6 ± 0.8 sessions and ASSISTED 12.5 ± 0.7.

Based on sprint trial 1 and 2 from the pre-training tests, typical error (TE), coefficient of variation (CV%), and intraclass correlation coefficients (ICC) with 95% confidence intervals were 0.03 s (0.01–0.04) 2.00% (1.34–2.65), and 0.84 (0.36–0.76) for 0–5-m sprint time; 0.05 s (0.04–0.08), 0.99% (0.39–1.58), and 0.92 (0.77–0.98) for 0-20-m sprint time; 0.15 m/s (0.11–0.22), 1.45% (0.99–1.89), and 0.93 (0.83–0.97) for *v0*; 0.25 N kg^−1^, 3.02% (1.82–4.23), and 0.87 (0.69–0.95) for F0 N kg^−1^; and 0.03 (N.s.m^−1^ kg^−1^), 5.59% and 0.91 (0.79–0.96) for sprint FV profile, respectively. 

Inferential statistics on athletes completing the study are presented in Table 2. Correlations and tapering results in Figure 2 and Figure 3, respectively. 

Although both groups showed moderate effects for change in the sprint FV profile (RESISTED: ES: −0.86, p = 0.29, ASSISTED: ES: −0.60, p = 0.28), large within-group response variation was present in both groups (RESISTED: 9.00% ± 15.33, ASSISTED: 5.17% ± 17.56), as shown in Figure 2 and Figure 3. Both groups showed a significant very large to nearly perfect correlation between initial sprint FV profile and % change in sprint FV profile (RESISTED: r = −0.95, p < 0.01, ASSISTED: r = −0.79, p < 0.01). This included negative responders in both groups (RESISTED: 5/6 (1 < MDC%), ASSISTED: = 5/10 (2 < MDC%) see Figure 2). Between-group post-hoc analysis with controlling for initial values (ANCOVA) revealed significant differences in F*0* (p = 0.02, ES: 0.74) and the sprint FV profile (p = 0.02, ES: 0.86). No other significant changes were found. Within-group t-test post-hoc analysis showed significant improvements in 20-m split time performance in RESISTED group with a large effect size (p = 0.007, ES = −1.23). No other significant changes were found, including body mass (p > 0.05, ES < 0.21).

## 4. Discussion

The main finding of this study is that the initial sprint FV profile is a useful tool to explain the degree of potential for creating changes within the sprint FV profile with horizontally oriented modalities. The results demonstrate the importance of accounting for the individual’s initial mechanical status to potentially avoid large within-group response variation. Therefore, to some degree, these results may explain the contrasting or unclear results of previous studies and within this study. Specifically, the athletes’ initial sprint mechanical properties have not been considered during group allocation, and the resistance has not been standardized according to a specific velocity [7,8]. 

To clarify, the heavy sled training within the RESISTED group had the primary aim of increasing the force-orientation of the sprint FV profile and associated improvements in early acceleration. In constrast, the primary aim of the assistance training in the ASSISTED group was to create more velocity-oriented profiles by improving horizontal force toward the other end of the FV spectrum (thus fixating on *v0*). On a group level, both groups managed to shift the FV profiles in the opposite direction from each other, leading to a significant between-group difference. This corresponded to a moderate effect on shifting the sprint FV profile in the desired direction (Figure 2). However, it is interesting to consider the underlying reasons why the RESISTED group had the only significant within-group performance improvement (20-m time), the only between-group performance improvement (F*0*), and a stronger correlation between sprint FV profile changes and the initial FV profile. Based on the specificity of assisted training, this was a logical result. Compared to heavy resisted training the window for specific neurophysiological changes is likely much narrower [16,30]. The results presented in Figure 2 demonstrate this indirectly. Within the RESISTED group, 83% responded in the desired direction (5/6: Figure 1A), whereas, in the ASSISTED group, only 50% responded in the desired direction (5/10: Figure 2B). Although the individual should be considered whenever possible, on a group level, our results correspond with previous literature, showing that training at the upper end of the FV spectrum is more likely to transfer to power output [30]. This is likely because high force output training more often provides a larger range of neurophysiological stimuli [12]. Another reason for the statistical differences within the groups could the possible suboptimal programming strategies used for the assisted training, such as less time under tension and degree of assistance. Although there was an effort to counterbalance time under tension (10-m vs. 20-m sprints), the resisted efforts still lasted roughly 40% longer. This could partly explain why maximal velocity was not further improved compared to the RESISTED group. As the target was to focus on the FV spectrums high-velocity range, this could have been improved by increasing the distance from 20 m to 30 m. However, as there is little data available on the risks of overspeed, a shorter distance was chosen for safety concerns.

As negative responders were those with values at the end of the range of their respective groups (i.e., the highest force and velocity orientation for resisted and assisted, respectively), this may demonstrate the potential value in planning sprint training stimuli in accordance with the individual’s initial sprint FV profile. Our results might indicate a possible abstract cutoff zone for the prescription of training load/modalities. Specifically, sprint FV profile slopes approximating below −0.92 may not respond to heavy resisted training, and FV profiles above −0.83 may not respond to assisted training (Figure 2). Despite the small sample size, our unpublished data, currently in review from another high-level cohort completing heavy resisted sprinting, indicate a highly similar cutoff zone. It is important also to mention that an individualized training approach may even have a positive influence on injury risk management, as overuse injuries can be caused by overly focusing on strengths instead of weaknesses. 

It is important to state that a desired profile orientation change could take place in an unwanted manner by negatively influencing the opposite end of the FV spectrum. For example, increased force-orientation can also be achieved by decreasing horizontal capabilities around maximal velocity, therefore leading to a reduction in *v0* while *F0* is maintained. The main difference is that this type of change would not be accommodated with a desired performance response. The responders in the RESISTED group (5/6) were able to dominantly change their profile coherently via increases in *F0* (6–22%) instead of lowering *v0*, albeit 2/6 were under the MDC% threshold (Figure 3A). This also corresponded with improvements in acceleration (20-m split times, p = 0.007). Furthermore, *F0* response was nearly perfectly associated with initial values of *F0* within both groups (RESISTED: r = −0.98, ASSISTED: r = −0.83, p < 0.01). This demonstrates that at least within this population, initial values of *F0* and the initial sprint FV profile are equally valuable to predict training outcome. It is unclear whether the initial sprint FV profile becomes more valuable at a higher sprint performance level. Out of the five responders in the ASSISTED group that improved their velocity orientation, three did not respond in an ideal manner. This was because they dominantly changed their sprint FV profile by lowering their early acceleration capabilities instead of increasing *v0* (Figure 3), therefore becoming more velocity-oriented without increasing performance. Based on these results, the utility and accuracy of different formats of assisted training to create more ideal changes in sprint FV profiles needs more clarification.

Whereas more studies are needed to clarify whether certain modalities can accelerate and/or even boost training response, our results already demonstrate some more accommodating solutions. For example, athletes within the ASSISTED group that had sprint FV profiles approximating −0.83 or lower became more force-oriented, leading to improved early acceleration performance. This means that professional rugby athletes with a large sprint FV profile velocity orientation may effectively become more force oriented by simply increasing short distance sprint volume. 

The response to tapering post-intervention was varied between individuals (Figure 3), with most athletes reaching their peak performance (maximal force and velocity) within the two first weeks, corresponding with previous literature [18]. In both groups, force orientation capacity within the sprint FV profile seemed to reduce how far the taper went, although this more evident in the RESISTED group. This was explained more by early acceleration capability decreasing with time, which was more evident in the RESISTED group, while maximal velocity capacity slightly increased in both groups with the taper weeks, but not past the MDC. This is likely because a complete taper of sled training was not optimal, but the maximal velocity capacity remained high due to the taper included 2 × 30-m maximal sprints once a week. 

## 5. Conclusions

Using a combined force-velocity and load-velocity method to select loading in sprint training could be useful for coaches aiming to improve individualization within rugby populations. Specifically, when acting upon the individuals sprint FV profile data, heavy resisted and assisted training seem to be useful training tools when appropriately timed in-season. However, free sprinting should not be neglected and should be replaced with careful consideration. These results also demonstrate that sprint performance and mechanical outputs are sensitive to taper in team sport after an individualization training based on sprint FV profiling. An interesting question for future studies will be what the “optimal” sprint FV profile for maximizing sprint performance is and what combination of loads are most optimal to aid progress for a specific athlete within a specific sport. No studies have yet proposed a theoretical base for this. More studies focusing on the individual training adaptations are needed to better understand the appropriate loading parameters, both from macroscopic (improvements in mechanical variables) and microscopic (athlete running technique variables) viewpoints. Nevertheless, our results provide initial evidence that taking individual sprint FV profiles into account is important when deciding what type of horizontally oriented training is used to influence sprint performance within professional rugby.

## Figures and Tables

**Figure 1 sports-08-00074-f001:**
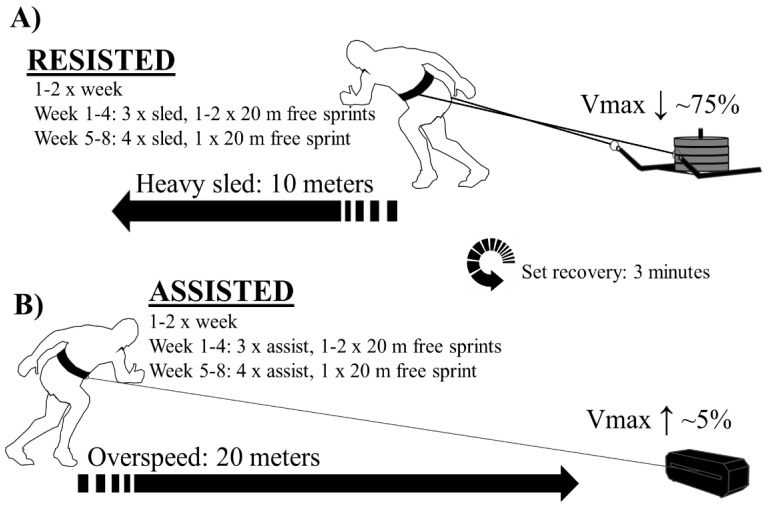
Training intervention design.

**Figure 2 sports-08-00074-f002:**
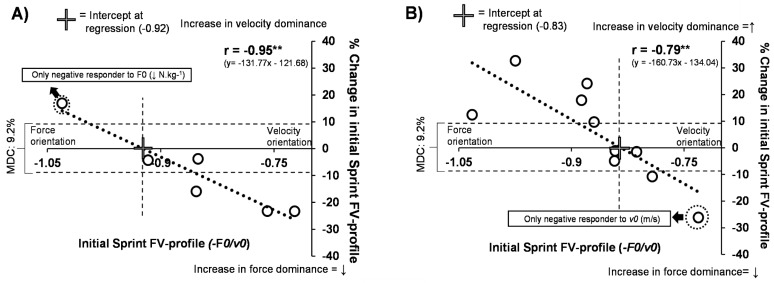
Correlations between initial and changes in Sprint FV profile. Correlations in RESISTED and ASSISTED training groups. Figure A and B demonstrate the group correlations and trendlines between the initial Sprint FV profile value and % changes within the profile. For the RESISTED group (**A**), the resisted sprint training had the desired effect on the profile in the athletes that had Sprint FV profiles < −0.92. For the ASSISTED group (**B**), assisted training had the desired effect on the profile in the athletes that had sprint FV profiles > −0.83. Values are MDC%, Minimal detectable change (calculated from between week % fluctuations in variable within season) kg, kilogram; v0, maximal theoretical running velocity; s, second; F0, maximal theoretical horizontal force; N, newton. * p < 0.05, ** p < 0.01.

**Figure 3 sports-08-00074-f003:**
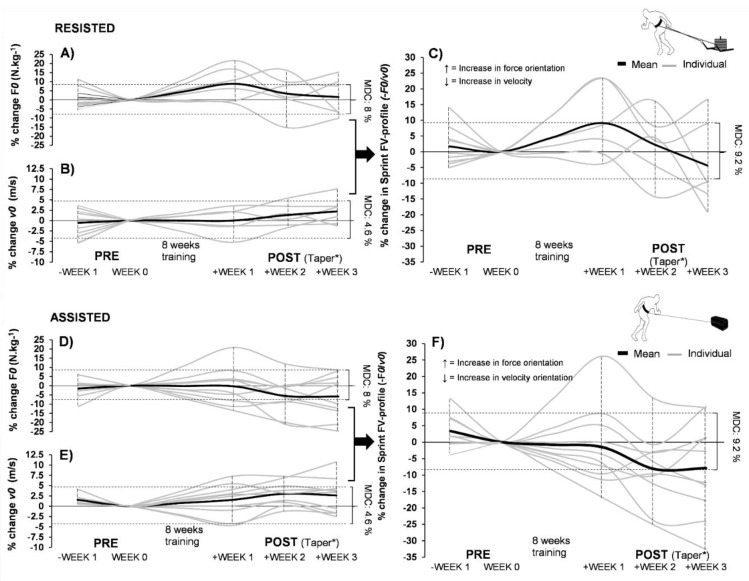
The sprint FV profile and its component changes before and after the intervention. Normal weekly fluctuations and tapering kinetics for the sprint FV profile and its respective components within RESISTED and ASSISTED training groups. Sub-graphs **A**), **B**) (RESISTED), **D**), **E**) (ASSISTED) demonstrate the group trends and individual data for tapering in both ends of the sprint FV profile: F*0* and V*0*, within season after an eight-week protocol. Sub-graph **C**) (RESISTED) and **F**) (ASSITED) demonstrate the tapering of the sprint FV profile. Dropouts are also presented in terms of their PRE measurement values to provide a better representation of initial within-group heterogeneity and between-group homogeneity. Values: kg, kilogram; v0, maximal theoretical running velocity; s, second; F0, maximal theoretical horizontal force; N, newton. * = Tapering method used: Full cessation of sled and assisted sprinting, 2 x 2 20-m sprints per each tapering week.

**Table 1 sports-08-00074-t001:** Participant information and weekly programming structure during study.

Group		N	Age (y)		Height (m)	Body Mass (kg)		Initial FV profile (-*F0/v0*)		load used as % of BM		Velocity change (%)		
RESISTED		6	19.3 ± 0.30		1.83 ± 0.10	91.4 ± 15.3		−0.85 ± 0.08		1.02 ± 0.05		−75.0 ± 6.31 *		
ASSISTED		10	20.1 ± 1.00		1.90 ± 0.10	94.4 ± 9.10		−0.86 ± 0.09		−0.05 ± 0.01		+6.48 ± 2.00		
Weekly programming structure for RESISTED and ASSISTED groups.														
Day	Week −1	Week 0	Week 1	Week 2	Week 3	Week 4	Week 5		Week 6	Week 7	Week 8	Week +1	Week +2	Week +3
MON	UB/TEC	UB/TEC	UB/TEC	UB/TEC	UB/TEC	UB/TEC	UB		UB/TEC	UB/TEC	UB/TEC	UB/TEC	UB/TEC	UB/TEC
TUE	AS or RE **/LB/Tec Tac	AS or RE ***/LB/Tec Tac	AS or RE/LB/Tec Tac	AS or RE/LB/Tec Tac	AS or RE/LB/Tec Tac	AS or RE/LB/Tec Tac	AS or RE/LB/Tec Tac		AS or RE/LB/Tec Tac	AS or RE/LB/Tec Tac	AS or RE/LB/Tec Tac	S/LB/Tec Tac	S/LB/Tec Tac	S/LB/Tec Tac
THU	MB/Tec Tac	MB/Tec Tac	AS or RE/MB/Tec Tac	AS or RE/MB/Tec Tac	AS or RE/MB/Tec Tac	AS or RE/MB/Tec Tac	AS or RE/MB		AS or RE/MB/Tec Tac	AS or RE/MB/Tec Tac	AS or RE/MB/Tec Tac	MB/Tec Tac	MB/Tec Tac	MB/Tec Tac
FRI	Tac	Tac	Tac	Tac	Tac	Tac	CG		Tac	Tac	Tac	Tac	CG	Tac
SAT	M	M	M	M	M	M	OFF		M	M	M	M	OFF	M
Load *	2000	2430	2860	1880	2240	2150	1440		2620	1910	2580	1510	1370	1450

Values are kg, kilogram; Y, years; BM, body mass; v*0*, maximal theoretical velocity; s, second; F*0*, maximal theoretical horizontal force; N, newton. *, Due to not being able to verify sled velocity in training, the CV% (coefficient of variation) was calculated based on pilot data to indicate realistic % error of prediction model (similar sled and surface); Load, Rated Perceived Exertion x Training Duration for week; TEC, Technical Training; TAC, Tactical Training; UB, Upper Body Training; LB, Lower Body; MB, Mixed full body training; CG, Conditioning Games; AS, Assisted sprint training; RE, Resisted sprint training; S, Sprint training and testing; M, Match; **, Familiarization week 1; ***, Familiarization and testing week 2.

**Table 2 sports-08-00074-t002:** Inferential statistics for within- and between-group comparisons.

RESISTED and ASSISTED Within-Group Inferential Statistics								Between-Group Differences (ANCOVA)	
Variable	Group	Homogeneity of variance (Levene’s test)	Pre	Post	Post -Pre			Post -Pre	
			x̅ ± SD	x̅ ± SD	%∆ ± SD	ES; ±95% CL	p value	ES; ±95% CL	p value
*v0* (m/s)	RESISTED	F(1, 19) = 0.2692 p = 0.59	8.75 ± 0.47	9.08 ± 0.48	3.21 ± 2.37	0.70 (−0.34–1.74)	0.02	0.06 (−0.95–1.07), p = 0.81 #	
	ASSISTED		8.71 ± 0.50	8.94 ± 0.46	3.40 ± 4.15	0.47 (−0.38–1.32)	0.03		
F0 (N kg^−1^)	RESISTED	F(1, 19) = 1.669 p = 0.21	7.46 ± 0.73	8.08 ± 0.26	8.95 ± 13.2	1.13 (0.02–2.24)	0.22	−0.74 (−1.78–0.31), p = 0.02	
	ASSISTED		7.46 ± 0.41	7.32 ± 0.78	−1.27 ±14.4	−0.23 (−1.10–0.66)	0.69		
Pmax (W kg^−1^)	RESISTED	F(1, 19) = 0.041 p = 0.84	16.1 ± 1.15	17.7 ± 0.78	9.21 ± 12.0	1.58 (0.40–2.76)	0.17	−0.55 (−1.58–0.48), p = 0.16	
	ASSISTED		16.1 ± 1.50	16.4 ± 1.82	2.75 ± 11.3	0.15 (−0.69–0.99)	0.51		
Sprint FV Profile (-F*0*/*v0*)	RESISTED	F(1, 19) = 0.474 p = 0.49	−0.85 ± 0.08	−0.92 ± 0.05	−9.00 ±15.3	−1.01 (−2.11–0.08)	0.29	0.86 (−0.20–1.91), p = 0.02	
	ASSISTED		−0.86 ± 0.09	−0.81 ± 0.08	5.17 ± 17.6	−0.60 (−1.50–0.29)	0.27		
5-meter time (s)	RESISTED	F(1, 19) = 1.284 p = 0.27	1.38 ± 0.05	1.33 ± 0.02	−3.22 ± 4.62	−1.19 (−2.28–0.10)	0.14	0.74 (−0.30–1.79), p = 0.07	
	ASSISTED		1.38 ± 0.03	1.40 ± 0.07	0.68 ± 5.80	0.23 (−0.65–−1.11)	0.64		
20-meter time (s)	RESISTED	F(1, 19) = 0.780, p = 0.38	3.44 ± 0.11	3.32 ± 0.08	−3.25 ± 1.78	−1.23 (−2.33–0.14)	0.007 *	1.16 (0.07–2.25), p = 0.07 #	
	ASSISTED		3.45 ± 0.08	3.45 ± 0.37	0.23 ± 3.86	0.02 (−0.82–0.86)	0.91		

Values are mean ± standard deviation, percent change ± standard deviation and standardized effect size; ±95% confidence limits. Abbreviations: n, sample size; x̅, mean; SD, standard deviation, %Δ, percent change; ES, effect size; 95% CL, 95% confidence limits; kg, kilogram; v0, maximal theoretical running velocity; s, second; F*0*, maximal theoretical horizontal force; N, newton; ES, effect size - small (≥ 0.2), moderate (≥ 0.6), large (≥ 1.2) effects. * p < 0.008 (Bonferroni post-hoc correction). #, covariate reached significance (p < 0.05).

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
