# Peer review of "Individual Sprint Force-Velocity Profile Adaptations to In-Season Assisted and Resisted Velocity-Based Training in Professional Rugby"

_sports, 2020, doi:10.3390/sports8050074_

Round 1
Reviewer 1 Report
This interesting study evaluated the correlation between an initial sprint FV-profile and the change in sprint FV-profile by performing either resisted or assisted based sprint programs over 8 weeks of training and a 3 week taper. The main results the authors report is a significant correlation between the initial sprint FV-profile and the change in sprint FV-profile (r=-0.95 for resisted and r=-0.79 for assisted).
From what I understand about the study methodology testing procedures and the results, it is noted that a 20m sprint performance improvement was made in the resisted training group only (and not the assisted training group). I am wondering if the authors considered testing the athletes for sprint performance over a longer distance (say 40-60m) to see if the assisted training group would have seen a significant increase in sprint performance over the longer distance? It may be argued that resisted sprint training may enhance acceleration capabilities more so than assisted sprint training which may have more of an influence once maximal velocity is reached. Due to the shorter nature of the 20m sprint test, it would make sense that the resisted group would improve their performance in this shorter distance that requires maximal acceleration whereas a longer sprint, of say 40-60m, would likely see improvements if the athletes worked on assisted sprinting. The authors may want to mention this in their discussion.
Minor comments:
Line 31 - define FV the first time used in the abstract.
Line 38 - how many athletes finished the entire study i.e. N=15 of N=16? There is discrepancy in the reported results. In Figure 2, there is N=6 in the resisted group and N=9 in the assisted group for the individual data points which would make the N=15 not 16. Please clarify.
Line 103 - sentence structure confusing.
Table 1 - should it state "...structure for RESISTED and ASSISTED groups"? Currently it states "RESISTED and RESISTED groups". Also, shouldn't the AS&RE acronym be AS or RE as the groups performed differing interventions?
Line 258 - explain what TE is referring to the first time the acronym is used.
Lines 275-286 - please clarify the number of athletes per group as it is confusing in this current explanation of the athletes finishing the interventions.
Figure 3 - currently it is very difficult to see the individual athlete data points in this figure. Also Fig 3 legend line 319 individual is spelled incorrectly.
Line 331-332 - confusing sentence structure; please clarify.
Line 393 - confusing sentence structure; please clarify.
Author Response
Reviewer comment 1: This interesting study evaluated the correlation between an initial sprint FV-profile and the change in sprint FV-profile by performing either resisted or assisted based sprint programs over 8 weeks of training and a 3 week taper. The main results the authors report is a significant correlation between the initial sprint FV-profile and the change in sprint FV-profile (r=-0.95 for resisted and r=-0.79 for assisted).
From what I understand about the study methodology testing procedures and the results, it is noted that a 20m sprint performance improvement was made in the resisted training group only (and not the assisted training group). I am wondering if the authors considered testing the athletes for sprint performance over a longer distance (say 40-60m) to see if the assisted training group would have seen a significant increase in sprint performance over the longer distance? It may be argued that resisted sprint training may enhance acceleration capabilities more so than assisted sprint training which may have more of an influence once maximal velocity is reached. Due to the shorter nature of the 20m sprint test, it would make sense that the resisted group would improve their performance in this shorter distance that requires maximal acceleration whereas a longer sprint, of say 40-60m, would likely see improvements if the athletes worked on assisted sprinting. The authors may want to mention this in their discussion.
Response: Thank you for your comment. It is important to note that this 20 m improvement was only a within group change, so its not strong evidence in this type of study set up (no control group, small sample size etc). It is an interesting question about the longer distances. In general (excluding rugby sevens) sprints longer than 20-30 meters are not normal in rugby and most athletes reach max velocity already at 25 meters (in our cohort at least). The assisted sprinting was mostly about providing a target stimuli at an upright position. The athlete is upright at about 10 meters, so about 50% of the sprint is done in a maximal velocity type position. As both groups improved maximal velocity, it would probably have affected split times at some point, so 40-60 m etc. But this was not our focus and as stated not important for the sport. Our target was whether the athlete could increase maximal velocity within 30 meters, thus higher force production at higher speeds. If we would have used a rolling start and focused on getting in 20-40 meters at a peak velocity, then 40-60 split times would be worth testing.
However, in hindsight, the volume/time under tension provided to the overspeed group vs the resisted group was in our opinion the biggest limitation. The time under tension difference is still quite large even though we tried to account for it with using only 10-meter sprints in the resisted group. One solution would have been to have the assisted group complete 1-2 extra sprints per session. Considering the target of stimulating the late end of the acceleration curve, another likely better option would have been to use 30-meter sprints in the overspeed group instead of 20. But we were worried about the injury risk increasing in-season so we stuck to 20 meters. We hope another research group tries out longer distances.
We have now added this in more detail as a limitation into the text on line 405:
“Although there was an effort to counter-balance time under tension (10-m vs 20-m sprints), the resisted efforts still lasted roughly 40% longer. This could partly explain why maximal velocity was not further improved compared to the RESISTED group. As the target was to focus on the FV-spectrums high-velocity range, this could have been improved by increasing the distance from 20 to 30 meters. However, as there is little data available on the risks of overspeed, a shorter distance was chosen for safety concerns.”
Minor comments:
Reviewer comment 2: Line 31 - define FV the first time used in the abstract.
Response: Corrected thank you (Line 30 now)
Reviewer comment 3: Line 38 - how many athletes finished the entire study i.e. N=15 of N=16? There is discrepancy in the reported results. In Figure 2, there is N=6 in the resisted group and N=9 in the assisted group for the individual data points which would make the N=15 not 16. Please clarify.
Response: Thank you. 16 finished the study as stated in the abstract and we accidently wrote that there were 6 dropouts but actually there were only 5. This has been corrected (now line 128). We understand that one of the data points is difficult to see, as it was partly behind the x axis “-0.85” mark. We have now scaled the x axis differently in the figure to make it visible, thank you for the comment.
Reviewer comment 4: Line 103 - sentence structure confusing.
Response: We agree that the sentence is too long, which makes it hard to follow. We have made it into two separate sentences that read (Now line 102 – 105):
“Furthermore, optimizing the length between training completion and post testing is of additional interest. It is common that post testing is done the week after program completion, even though it has been shown both on a group- and individual level that peaking response can be further delayed [18].”
Reviewer comment 5: Table 1 - should it state "...structure for RESISTED and ASSISTED groups"? Currently it states "RESISTED and RESISTED groups". Also, shouldn't the AS&RE acronym be AS or RE as the groups performed differing interventions?
Response: Table 1 has been corrected starting on line 146, thank you. “AS&RE” has been changes to “AS or RE”
Reviewer comment 6: Line 258 - explain what TE is referring to the first time the acronym is used.
Response: Corrected on line 280, thank you.
Reviewer comment 7: Lines 275-286 - please clarify the number of athletes per group as it is confusing in this current explanation of the athletes finishing the interventions.
Response: Another reviewer also asked for improvements in this area (they wanted slightly more details). Hence we combined your request with theirs and find the paragraph much better now, it has also been moved to the participant section instead of the results due to reviewer request on lines 128 – 139:
““Due to scheduling and injury issues, five out of 21 athletes could not perform the minimum required sessions and/or the post-testing and were removed from the study. Specifically, three of the dropouts were due to scheduling issues (international games, RESISTED N = 1, ASSISTED N = 2) that led to the players missing the post testing. Two athletes dropped out due to injuries (RESISTED N = 2). According to the team physiotherapist, one was a contact injury sustained in a match and the other was a reaggravated chronic foot problem reported in the first weeks of training possibly aggravated by the sled protocol. Consequently, the final analysis was performed on 16 athletes: 6 athletes completing heavy resisted training training (age: 19±0.3 years; height: 1.83 ± 0.1 m; mass: 91.4 ± 15.3 kg) and 10 athletes performing the assisted training (age: 20 ± 1 years; height: 1.9 ± 0.1 m; mass: 94.4 ± 9.1 kg).”
Reviewer comment 8: Figure 3 - currently it is very difficult to see the individual athlete data points in this figure. Also Fig 3 legend line 319 individual is spelled incorrectly.
Response: Thank you. We have corrected the spelling mistake and made the individual lines thicker.
Reviewer comment 9: Line 331-332 - confusing sentence structure; please clarify.
Response: This sentence has now been changed, thank you, now on line 379 – 381:
“Specifically, the athletes initial sprint mechanical properties have not been considered during group allocation, in combination with not standardizing the resistance according to a specific velocity [7,8].”
Reviewer comment 10: Line 393 - confusing sentence structure; please clarify.
Reviewer comment 1:
Response: This section has now been improved, thank you. This has been done on line 448 - 455:
“While more studies are needed in clarifying whether certain modalities can accelerate and/or even boost training response, our results already demonstrate some more accommodating solutions. For example, athletes within the ASSISTED group that had sprint FV-profiles approximating -0.83 or lower became more force oriented leading to improved early acceleration performance. This means that professional rugby athletes with a large sprint FV-profile velocity orientation may effectively become more force oriented by simply increasing short distance sprint volume. “
Reviewer 2 Report
ABSTRACT
line 40: remove "very large to nearly perfect"
STUDY DESIGN
line 160: remove all "unpublished observations" you discuss throughout this entire section. that type of reference illegitimizes your design, findings, etc. add more references to literature, avoid using "unpublished observations." if there is something specific you need to modify within your own testing space/equipment, reference how it is modified based on previously approved and validated protocol.
this appears in a few other locations within this section. revise accordingly.
GROUP ALLOCATION
line 184: please clarify "due to coaching staff preference." why were two additional athletes intentionally added to the assisted group?
RESULTS
line 277-280: the injury issues need more elaboration. the "contact injury" is reported to have been sustained in a match, but perhaps the overall overuse of the training, combined with match play led to injury. additionally, "possibly aggravated by the sled protocol" sounds inconsistent and unsure. was it aggravated by the protocol or not? what did the participant report as reason for dropping out? lastly, describe the scheduling issues that were reported. how much of the protocol did these participants complete? when did they drop out? were their performance values trending in any certain direction?
Author Response
Reviewer comment 1:
ABSTRACT
line 40: remove "very large to nearly perfect"
Response: This has been removed, thank you
STUDY DESIGN
Reviewer comment 2: line 160: remove all "unpublished observations" you discuss throughout this entire section. that type of reference illegitimizes your design, findings, etc. add more references to literature, avoid using "unpublished observations." if there is something specific you need to modify within your own testing space/equipment, reference how it is modified based on previously approved and validated protocol.
this appears in a few other locations within this section. revise accordingly.
Response: Thank you for your comment. We removed the word “unpublished” from line 160 (now line 173), but kept the word “observation” throughout the paper. We thus disagree that stating if some ideas are based on observations illegitimizes our design. Our aim was to be transparent in how we came to these conclusions. For example, there are no publications available using the degree of resistance we used in our study. Hence, we had to run some pilots and based on observations from these pilots we determined what is safe and plausible. This should be mentioned in our opinion. However, we cited the studies as close as possible to the observations also in each instance. Therefore, we hope you understand why we want to keep this wording.
GROUP ALLOCATION
Reviewer comment 3: line 184: please clarify "due to coaching staff preference." why were two additional athletes intentionally added to the assisted group?
Response: As you know, coaches have different relationships with different athletes depending on their experience level and personality traits. Concerning, these two athletes, they felt that based on their personalities, they don’t think they will stick to the protocol with the required motivation if they are not in the overspeed group. As this was a study within a professional team setting, we had no say in this choice. We feel that it is unnecessary to add details on “personality traits” and “relationships between coaching staff and athletes” into this paper. We however can add that it was both an athlete and staff preference. Therefore, we have made the following correction to line 196 - 198:
“Athletes were allocated to both groups to balance variance, with 2 additional athletes allocated to the ASSISTED group due to athlete and coaching staff preference”
RESULTS
Reviewer comment 4: line 277-280: the injury issues need more elaboration. the "contact injury" is reported to have been sustained in a match, but perhaps the overall overuse of the training, combined with match play led to injury. additionally, "possibly aggravated by the sled protocol" sounds inconsistent and unsure. was it aggravated by the protocol or not? what did the participant report as reason for dropping out? lastly, describe the scheduling issues that were reported. how much of the protocol did these participants complete? when did they drop out? were their performance values trending in any certain direction?
Response: Thank you for your comment. It was important for us to be transparent. However, we want to avoid speculation also as injury pathology is not always as simple as yes or no when you don’t have all the data. We agree that contact injuries can be possibly hindered or reduced in how serious they are based on the status (physical, mental) of the athlete. However, as this was a contact injury instead of a non-contact injury in a sport where contact injuries are highly normal, it should be enough information if the physio didn’t elaborate further. Which brings us to the fact that it is highly important to mention that we asked the team physiotherapist on what their opinion is concerning the pathology of both injuries. The contact injury was just explained by bad luck (apparently a sketchy tackle by an opposing team player, no fatigue was mentioned) and the second injury (the chronic foot problem) the physio stated that they were quite unsure it had anything to do with the sled training as it happened in the first weeks, it was a repeating problem with the athlete, and no other foot problems have been reported from the sled training at any timepoint. We have now added to the text that it was a reaggravated chronic injury happening in the first weeks of training. As you know, injury pathology is not always black and white. However, we wanted to remain transparent and not seem that we are trying to hide something, as it was still a possibility with the higher loading used in our study. Thus the word “possible” is placed as nobody knows for sure and if the reader is confused its because we cannot give a straight answer. Giving such an answer would be highly difficult in any high level setting, as it would require highly accurate impact data of foot strikes (maybe using IMU) to get an idea if there have been inappropriate spikes in impact to a specific tissue. We have however added the important statement that the injury backgrounds were determined by a professional; the team physiotherapist.
We have also described the scheduling issues of the 3 dropouts. Furthermore, the section is now moved to the participant section due to reviewer request on lines 128 – 135:
“Due to scheduling and injury issues, five out of 21 athletes could not perform the minimum required sessions and/or the post-testing and were removed from the study. Specifically, three of the dropouts were due to scheduling issues (international games, RESISTED N = 1, ASSISTED N = 2) that led to the players missing the post testing. Two athletes dropped out due to injuries (RESISTED N = 2). According to the team physiotherapist, one was a contact injury sustained in a match and the other was a reaggravated chronic foot problem reported in the first weeks of training possibly aggravated by the sled protocol.”
Reviewer 3 Report
Dear authors,
Thank you so much to allow me to review the manuscript entitled “Individual adaptations to assisted and resisted velocity-based sprint training in professional rugby athletes in-season: exploring both ends of the force-velocity spectrum”, which aimed to quantify the influence of initial sprint FV-properties on changes in individual sprint FV-profiles using training at different ends of the sprint FV-spectrum and to demonstrate individual peaking responses. I enjoyed reading it and I think that the manuscript could has a great impact on neuromuscular training knowledge, specifically in a hot topic in sport sciences. Nevertheless, I have some comments:
- Title is focused on “Individual adaptations to assisted and resisted velocity-based sprint training 2 in professional rugby athletes in-season”. However, hypothesis and conclusions are focused towards the influence of initial sprint FV-properties on changes in individual sprint FV-profiles. I believe that authors should homogenize title and hypothesis-conclusions.
- The abstract should end with a specific practical application.
- Introduction is well-structured and well-written. However, it would be interesting to clarify the term “opposite ends sprint FV-spectrum” to improve readers' understanding.
- In methods section (participants), inclusion/exclusion criteria must be declared, even more so when 5 athletes were not included in the final analysis.
- Following with the previous comment, you must include the reasons why the 5 athletes were ignored.
- Information of athletes that could not perform the minimum required sessions and/or the post-testing and were removed from the study must be included in the methods section. In this regard, all the information, tables and figures must be referring to the final sample (6 vs 10).
- Authors declared that a convenience sample was used. Could you clarify it? Do you perform a power analysis?
- Authors used a two-tailed dependent and independent t-tests to examine within and between group differences. In this sense, I understand that when two variables were included: time (pre-post) and training program (resisted and assisted), an ANCOVA or a 2-way repeated measures analysis of variance (ANOVA) must be implemented to analyze between group differences. Could you explain and justify your selection?
- The manuscript must be presented in the specific template.
Author Response
Reviewer comment 1: Dear authors,
Thank you so much to allow me to review the manuscript entitled “Individual adaptations to assisted and resisted velocity-based sprint training in professional rugby athletes in-season: exploring both ends of the force-velocity spectrum”, which aimed to quantify the influence of initial sprint FV-properties on changes in individual sprint FV-profiles using training at different ends of the sprint FV-spectrum and to demonstrate individual peaking responses. I enjoyed reading it and I think that the manuscript could has a great impact on neuromuscular training knowledge, specifically in a hot topic in sport sciences. Nevertheless, I have some comments:
Response: Thank you for your kind words, we think your comments have helped to make the manuscript better.
Reviewer comment 2: Title is focused on “Individual adaptations to assisted and resisted velocity-based sprint training 2 in professional rugby athletes in-season”. However, hypothesis and conclusions are focused towards the influence of initial sprint FV-properties on changes in individual sprint FV-profiles. I believe that authors should homogenize title and hypothesis-conclusions.
This is a good point thank you. The manuscript focuses on the individual FV-profiles as a main outcome (abstract focus), and then the tapering and performance. We agree with you that the title should be changed to the main outcome, so we propose the following and hope you agree:
“Individual sprint force-velocity profile adaptations to assisted and resisted velocity-based sprint training in professional rugby.”
Reviewer comment 3: The abstract should end with a specific practical application.
Response: Thank you for your comment. As the abstract only allows 200 words, we cannot go into much detail. However, we feel that we have given a specific practical implication; take into account individual sprint fv-profiles to possibly improve training outcome, which is stated in other words in the last sentence:
Practitioners should consider utilizing the sprint FV-profile to improve the individual effectiveness of sprint training programs in high-level rugby athletes.
We could however add slightly more specific detail, in mentioning resisted and assisted training. Thus it has now been changed on line 44 to:
“Individual sprint force-velocity profile adaptations to in-season assisted and resisted velocity-based training in professional rugby.”
Reviewer comment 4: Introduction is well-structured and well-written. However, it would be interesting to clarify the term “opposite ends sprint FV-spectrum” to improve readers' understanding.
Response: Thank you for your comment. We feel that we have done this adequately on lines 84-91:
“In this regard, training using horizontal resistance, such as sleds, is used by practitioners to favor the development of the horizontal force component across the FV-spectrum. Thus, greater loads are used to aid development in force properties at low velocities and lighter loads at higher velocities, changing the FV-relationship with a bias towards force orientation [14–16]. On the other hand, assisted sprinting, such as using elastic cords, bungees, or a pulley for assistance, targets the opposite end of the spectrum and potentially increases velocity orientation of the FV-profile (improving the horizontal force component at high velocities) [16,17].”
Reviewer comment 5: In methods section (participants), inclusion/exclusion criteria must be declared, even more so when 5 athletes were not included in the final analysis.
Response: Thank you, we have now added the following sentence to the participants section on line 126:
“Inclusion criteria included being an active and healthy rugby player within the team, while exclusion criteria included completing less than 12 out of 16 training sessions and all testing sessions.”
Reviewer comment 6: Following with the previous comment, you must include the reasons why the 5 athletes were ignored.
Response: Thank you. They have all been mentioned in the same section due to your other request of moving this information to the methods instead of the results section, There have also been minor changes to that section based on other reviewer comments on lines on lines 128 – 139:
“Due to scheduling and injury issues, five athletes could not perform the minimum required sessions and/or the post-testing and were removed from the study. Specifically, three of the dropouts were due to scheduling issues (international games) that led to the players missing the post testing. Two athletes dropped out due to injuries. According to the team physiotherapist, one was a contact injury sustained in a match and the other was a reaggravated chronic foot problem reported in the first weeks of training possibly aggravated by the sled protocol. Consequently, the final analysis was performed on 16 athletes; 6 athletes completing heavy resisted training (age: 19±0.3 years; height: 1.83 ± 0.1 m; mass: 91.4 ± 15.3 kg) versus 10 athletes completing assisted training (age: 20 ± 1 years; height: 1.9 ± 0.1 m; mass: 94.4 ± 9.1 kg).”
Reviewer comment 7: Information of athletes that could not perform the minimum required sessions and/or the post-testing and were removed from the study must be included in the methods section. In this regard, all the information, tables and figures must be referring to the final sample (6 vs 10).
Response: Thank you for your comment. As mentioned in the comment above, this has been corrected and the entire paper now follows the 6 vs 10 design. We do mention though in the group allocation section how we started with a 9 vs 12 ratio and then moved to 6 vs 10 and this did not affect the homogeneity between groups (line 204 – 206).
Reviewer comment 8: Authors declared that a convenience sample was used. Could you clarify it? Do you perform a power analysis?
Response: No direct power analysis was performed. As you know its very difficult to get to study a population of elite rugby athletes hence it was a convenience sample based on contacts within our research team. As you are also aware, similar N=20 athlete population intervention studies are performed constantly in sports science to help improve our understanding. If you want us to refer to similar studies with similar sample size, please let us know and we will do it.
Reviewer comment 9: Authors used a two-tailed dependent and independent t-tests to examine within and between group differences. In this sense, I understand that when two variables were included: time (pre-post) and training program (resisted and assisted), an ANCOVA or a 2-way repeated measures analysis of variance (ANOVA) must be implemented to analyze between group differences. Could you explain and justify your selection? The manuscript must be presented in the specific template.
Response: Thank you for your response. We agree that a one-way ANCOVA is a smart choice, as we divided the groups based on a covariate (sprint FV-profiles) with one factor (Group). Hence, we have changed the between group analysis to this. This didn’t change the interpretation of the study results although new variables reached significance. F0 and FV-profiles reached significance by using the ANCOVA, which makes a lot of sense. Increased F0 represents force orientation and it was significantly improved in the RESISTED group, supporting the 20-m within-group performance changes. In terms of the sprint FV-profile, the target in this study was to move the FV-profile in opposite directions between the groups, so it makes sense that the FV-profiles reached a significant difference when controlling for initial values. These two results have been interpreted as follows in the following sections:
Results, line 307:
“Between-group post-hoc analysis with controlling for initial values (ANCOVA) revealed a significant differences in F0 (p = 0.02, ES: 0.74) and the sprint FV-profile (p = 0.02, ES: 0.86). No other significant changes were found.”
Also table 2 has been updated with the ANCOVA results.
Discussion, lines 387 – 392:
“On a group level, both groups managed to shift the FV-profiles in the opposite direction from each other, leading to a significant between-group difference. This corresponded to a moderate effect on shifting the sprint FV-profile in the desired direction (Figure 2). However, it is interesting to consider the underlying reasons why the RESISTED group had the only significant within-group performance improvement (20-m time), the only between-group performance improvement (F0), and a stronger correlation between sprint FV-profile changes and the initial FV-profile.”
Round 2
Reviewer 3 Report
The revised version of paper is written following my suggestions and is worthy for publication.